# An Improved Machine-Learning Approach for COVID-19 Prediction Using Harris Hawks Optimization and Feature Analysis Using SHAP

**DOI:** 10.3390/diagnostics12051023

**Published:** 2022-04-19

**Authors:** Kumar Debjit, Md Saiful Islam, Md. Abadur Rahman, Farhana Tazmim Pinki, Rajan Dev Nath, Saad Al-Ahmadi, Md. Shahadat Hossain, Khondoker Mirazul Mumenin, Md. Abdul Awal

**Affiliations:** 1Faculty of Health, Engineering and Sciences, University of Southern Queensland, 487-535 West Street, Toowoomba, QLD 4350, Australia; debjit_1@live.com; 2Department of Computer Science, College of Computer and Information Sciences, King Saud University, Riyadh 11543, Saudi Arabia; saislam@ksu.edu.sa (M.S.I.); salahmadi@ksu.edu.sa (S.A.-A.); 3Faculty of Science and Engineering, Southern Cross University, East Lismore, NSW 2480, Australia; md.abadur.rahman87@gmail.com; 4Computer Science and Engineering Discipline (CSE), Khulna University (KU), Khulna 9208, Bangladesh; farhana@ku.ac.bd; 5Faculty of Business, Education, Law and Arts, School of Commerce, University of Southern Queensland, Darling Heights, QLD 4350, Australia; rajangne1@gmail.com; 6Department of Quantitative Sciences, International University of Business Agriculture and Technology, Dhaka 1230, Bangladesh; shahadat_qs@iubat.edu; 7Electronics and Communication Engineering (ECE) Discipline, Khulna University (KU), Khulna 9208, Bangladesh; k.mirazulmumenin@gmail.com

**Keywords:** big COVID-19 data, HHO, machine learning, decision support system, healthcare

## Abstract

A healthcare monitoring system needs the support of recent technologies such as artificial intelligence (AI), machine learning (ML), and big data, especially during the COVID-19 pandemic. This global pandemic has already taken millions of lives. Both infected and uninfected people have generated big data where AI and ML can use to combat and detect COVID-19 at an early stage. Motivated by this, an improved ML framework for the early detection of this disease is proposed in this paper. The state-of-the-art Harris hawks optimization (HHO) algorithm with an improved objective function is proposed and applied to optimize the hyperparameters of the ML algorithms, namely HHO-based eXtreme gradient boosting (HHOXGB), light gradient boosting (HHOLGB), categorical boosting (HHOCAT), random forest (HHORF) and support vector classifier (HHOSVC). An ensemble technique was applied to these optimized ML models to improve the prediction performance. Our proposed method was applied to publicly available big COVID-19 data and yielded a prediction accuracy of 92.38% using the ensemble model. In contrast, HHOXGB provided the highest accuracy of 92.23% as a single optimized model. The performance of the proposed method was compared with the traditional algorithms and other ML-based methods. In both cases, our proposed method performed better. Furthermore, not only the classification improvement, but also the features are analyzed in terms of feature importance calculated by SHapely adaptive exPlanations (SHAP) values. A graphical user interface is also discussed as a potential tool for nonspecialist users such as clinical staff and nurses. The processed data, trained model, and codes related to this study are available at GitHub.

## 1. Introduction

The world is now facing a challenge to eradicate COVID-19 because of its continuous spreading [1]. People in Wuhan City, China, were the first to be affected by the virus in December 2019 [2,3]. Then it spread from person to person, creating the latest global health problem. The virus is known as SARS-CoV-2 [4], and the related disease is called COVID-19 [5].

The World Health Organization (WHO) declared the outbreak a public health emergency of international concern on 30 January 2020, and a pandemic on 11 March 2020. The virus causes illness, cough, fever, etc., and in severe cases, it may end up in severe pneumonia [6]. Pneumonia creates infection primarily within the air sacs of the lung, creating problems for the oxygen exchange [6].

The patient might need to be hospitalized to monitor symptoms. Sometimes, the patient must be kept in an intensive care unit (ICU), and in extreme cases, patients must be placed on a ventilator to help breathe properly [7]. COVID-19 is a serious disease due to its high community transmission and side effects [8]. The impact on the healthcare system is also high because of the large number of ICU patients and the need for long-term use of mechanical ventilators [9]. WHO also identified several new variants. Among them, the B.1.1.529 variant, called Omicron, has proven to be more contagious than other strains, leading to more people being hospitalized. In this situation, early diagnosis is essential for reducing the strain on the healthcare system. However, people remain vulnerable because of inadequate testing to detect COVID-19, especially in developing countries. At present, RT PCR is the most common diagnostic method used by hospitals worldwide to diagnose COVID-19. The most significant limitation of RT PCR is the high incidence of false negatives. For example, in China [10], RT PCR diagnostic tests give 41% false negatives. Another limitation of this method is that it takes two or more days or even a week to get results and creates a vulnerable situation in remote areas [11]. This extended time contributes to the spread of COVID-19 in the meantime; otherwise, the uninfected patients are kept isolated from others. To minimize this limitation, many research and development groups are actively working to find an effective method for proper diagnosing and treating the disease. This area of research includes medical, biotechnology and other fields such as data science, artificial intelligence and ML which can provide technical solutions to prevent and control this pandemic.

### 1.1. Related Works

Rajaraman and Antani [11] developed a deep-learning framework for COVID-19 detection using chest X-ray (CXR) data collected from COVID-affected people and then analyzed it using a convolution neural network (CNN) to evaluate the system. Yan et al. [12] proposed a model to detect severely affected COVID-19 patients using three clinical attributes and an ML algorithm. They used eXtreme gradient boosting (XGB) to predict their model. Another framework using ML and inpatient facility data for early detection of COVID-19 was presented by Awal et al. [13]. Here, Bayesian optimization was used to select optimal hyperparameters of the ML algorithms, and the adaptive synthetic (ADASYN) algorithm was applied to balance the dataset. The model showed 98.50% ACC using the XGB classifier. Here, the important features were also estimated using SHAP analysis, and a graphic interface was also presented for supporting medical staff. Kassania et al. [14] developed an ML-based COVID-19 detection system using CXR images where they applied a deep CNN to extract essential features. Two deep-learning techniques named DenseNet121 and ResNet50 were used for classification, and their accuracy was 99% and 98%, respectively. On the other hand, Saha et al. [15] proposed an automated diagnosis system for COVID-19 using CXR based on the ensemble method and CNN. Deep features extracted using CNN were utilized to combine four binary classifiers: support vector machine, random forest, AdaBoost and decision tree. Thus, they were able to build an ensemble of classifiers for better detection of COVID-19. The system showed better performance with 98.91% ACC. Rasheed et al. [16] presented the COVID-19 diagnosis system from a chest X-ray image using logistic regression (LR) and CNN. A principal component analysis (PCA) was used to reduce the dimension that increases the speed of the learning process. The CNN and LR showed 95.2–97.6% ACC without using PCA, whereas 97.6–100% ACC was obtained with PCA for COVID-19 detection.

### 1.2. Contributions

There are also many other methods for detecting COVID-19 where the ML algorithms are used, and CT (Chest) or CXR images are used as the input data. Generally, people need to go to the hospital to test for COVID-19, which is risky and time-consuming. Clinical data collected from home could be utilized in association with the ML-based systems for detecting COVID-19, where the inpatient facility data is used as the input dataset. The dataset contains fever, lack of smell and taste, cough, and chronic diseases such as diabetes, asthma, etc. Different boosting classifiers such as XGB, light gradient boosting (LGB), gradient boosting classifier (GBC), categorical boosting (CatBoost), random forest (RF), etc., could be used to train the model. Then these models could be combined to build ensemble classifiers that can increase performance. There are some hyperparameters in each classifier that can be optimized. Here we have used HHO to tune the hyperparameters, and it is expected that this model will be helpful and user-friendly for faster detection of COVID-19. The model’s performance could be evaluated using different metrics and graphic representations.

The main contributing part of this paper is described as follows:Since COVID-19 is a highly contagious disease, hospitals and diagnostic centers need extra precautions to test for COVID-19, ultimately increasing the costs and health hazards. The proposed model used the less expensive inpatient facility data which can be collected at home, instead of X-rays or CT scans to predict COVID-19 with the expectation of reducing patients’ visits to the hospital and diagnostic center.An ML framework is designed using HHO to detect COVID-19.The hyperparameters of the boosting classifiers are optimized using our method, and then the ensemble classifier has increased the performance of our model.The important features are estimated using SHapely adaptive exPlanations (SHAP) analysis.The performance is compared with other existing models.A decision support system and a clinically operable decision forest are created to support the medical staff.

The rest of the paper is organized as: Materials and Methods are described in Section 2. The experimental result is presented in Section 3. In Section 4, we present an organized discussion and comparison of our proposed system with other existing methods. The paper ends with a conclusion in Section 6.

## 2. Materials and Methods

### 2.1. Data Source

In this work, we used extensive data of COVID-19 patients gathered from a publicly available source (https://www.COVID19survivalcalculator.com/en/download, accessed on 23 February 2021). The master dataset contains data collected from 1,023,426 individuals with 98.80% COVID-19 negative and 1.20% COVID-19 positive patients. The dataset contains 59 columns about behavior, segmentation, geographic, health conditions, medications and risk values [17]. This dataset contains multiple columns such as *age*, *Body Mass Index (BMI)*, *sex*, alcohol, *cannabis*, *contacts_count*, *COVID19_symptoms*, *COVID19_result (positive and negative)*, *smoking*, different chronic diseases such as *asthma*, *kidney_disease*, *lung_disease*, *diabetes*, etc. Among the 59 columns in the master dataset, there are 40 categorical columns which are almost double the number of numeric columns. There are about 13.8% missing values and no duplicate value or row. It can be observed from the COVID-19 dataset that the variables, such as *region*, *income*, *insurance*, *immigrant*, *prescription_medicaton*, etc., have a higher rate of missing value than others. For this reason, the variables mentioned above have not been included in Table 1. Proposed system architecture please see in Figure 1.

### 2.2. Data Preprocessing

Firstly, metadata such as *survey_date*, *region*, *country*, *ip_latitude*, *ip_longitude*, *ip_accuracy*, *house_count*, *prescription_medication*, *opinion_infection*, *opinion_mortality*, *risk_infection*, *risk_mortality*, *nursing_home*, etc., have been removed as they are not relevant to our research objective. After that, the missing values of all attributes are imputed using iterative imputation techniques, where the BayesianRidge estimator has been used at each state of the iteration [18,19,20]. The sex and smoking attributes are encoded with a label encoder. Furthermore, *COVID-yes* is a minority class in the original dataset we used, whereas *COVID-no* is a majority class. Therefore, we have tried to make a nearly balanced dataset by taking a 1:3 ratio as *COVID-yes* and *COVID-no*, respectively, using the undersampling approach. The details of the dataset are shown in Table 1.

### 2.3. Classification Algorithms

Five classifiers, XGB, LGB, SVC and RF, have been applied in the proposed framework. These are the most recent classification algorithms used for COVID-19 classification [13]. We can evaluate the performance of these classification models with other existing works. Moreover, these classifiers have some hyperparameters that can be optimized using our proposed HHO-based framework. Finally, the SHAP analysis explains these classification algorithms to determine the most important clinical features for detecting COVID-19.

#### 2.3.1. XGB

The XGB model [21], associated with gradient boosting, combines weak classifiers to build a strong classifier in an iterative manner [22]. The residual is used to correct the previous predictor, and the loss function can be minimized at each iteration of gradient boosting. XGB adds regularization into the objective function to measure model performance [13].

The key hyperparameters of XGB are *learning_rate*, *max_depth*, *gamma*, *min_child_weight*, *colsample_by_tree*, *subsample* and *alpha*, where *learning_rate* controls the effect of adding new trees in order to prevent overfitting, *max_depth* limits a tree to the maximum number of nodes, *gamma* denotes loss reduction required for the next partition on a tree’s leaf node, *min_child_weight* means the minimum sum of weight needed, *colsample_by_tree* denotes the subsample ratio to build each tree, subsample builds a tree using the ratio of columns and *alpha* is the L1 regularization term on weight. By tuning these hyperparameters, the classification performance of the proposed system is increased.

#### 2.3.2. CatBoost

CatBoost, or categorical boosting, is an open-source ML algorithm developed by Yandex [1]. The “CatBoost” name comes from two words, “Category” and “Boosting” because CatBoost works with categorical features by converting categorical values into numbers. It can be used in ranking, recommendation systems, forecasting and and even personal assistants along with regression and classification. These advantages motivated us to choose CatBoost as the classifier for the proposed COVID-19 detection system.

The dominating hyperparameters of CatBoost are *depth*, *colsample_bylevel*, *subsample*, *n_estimator*, *learning_rate*, *l2_leaf_reg*, etc. The *learning_rate* and *subsample* have already been discussed in XGB. In addition, *Depth* represents the depth of the tree, *L2_leaf_reg* coefficient calculates the leaf value, *n_estimator* is the number of boosting iterations and *colsample_bylevel* is the ratio of columns for each level in tree building. These hyperparameters are adequately optimized to increase the performance of CatBoost.

#### 2.3.3. RF

RF combines different decision tree classifiers, where each classifier is obtained using a random vector from the input dataset. Each tree provides a unit vote for the target class to classify data [23]. The algorithm that is used for classification is given below.

Let *N* be the original dataset. A classification tree is constructed using Bootstrap. After random selection from the original dataset, the remaining samples create the out-of-bag data.At first, we have selected *n* variables randomly from each node of each tree. Then, a constant *m*
(m<n) is set, and we select *m* variables from *n* variables. After splitting the tree, the variable having the most classification ability is chosen from *m* variables based on the Gini index of the node impurity measurement. During classification, the threshold value of the variable is determined by checking each classification point. For a given training set *N*, we randomly selected one case and said it belongs to some class $C_{i}$.Every tree grows up to its maximum without any pruning.The classification outcome is obtained by the maximum voting result of the classifier.

The important hyperparameters of RF are *min_sample_split*, *max_depth*, *min_sample_leaf*, *n_estimator* and *criterion*. Here, *max_depth* is the maximum level of the tree, *min_sample_leaf* represents the minimum amount of samples that a node must keep after getting split, *min_sample_split* is the slightest amount of samples kept by an internal node required to split into further nodes, *n_estimator* controls the number of decision tree, and *criterion* measures the quality of splits. The optimization technique tunes the hyperparameters of RF and thus magnifies the performance of the classification task.

#### 2.3.4. LGB

In general, LGB, introduced by Ke et al. [24], relies on decision tree algorithms, and thus it produces a tree leaf-wise, while other algorithms grow depth-wise or level-wise (https://www.analyticsvidhya.com/blog/2017/06/which-algorithm-takes-the-crown-light-gbm-vs-xgboost/, accessed on 12 June 2017). The main advantages of LGB are that it can handle a large amount of data, has lower memory usage because of replacing the continuous value with discrete bins, can show better ACC than other boosting algorithms, and uses a histogram-based algorithm that makes the training process faster, etc.

The important hyperparameters of LGB are *learning_rate*, *number_of_leaves*, *feature_fraction*, *bagging_fraction*, *max_depth*, *max_bin*, *subsample*, *colsample_tree*, *min_child_samples* and *min_data _in_leaf*. Here, *learning_rate*, *max_depth*, *subsample* and *colsample_tree* are as previously discussed. In addition, *number_of_leaves* represents the number of leaves in the full tree, and the default value is 31; *feature_fraction* is considered in each iteration to set a fraction of the features; *bagging_fraction* helps execute bagging for getting faster results, *max_bin* is the maximum number of the bin; *min_child_samples* represent minimum number of data in one leaf; *min_data_in_leaf* is the lowest number of records of a leaf (https://www.analyticsvidhya.com/blog/2017/06/which-algorithm-takes-the-crown-light-gbm-vs-xgboost/, accessed on 12 June 2017). By optimizing these hyperparameters, the performance of LGB can be magnified.

#### 2.3.5. Ensemble Methods

An ensemble method is the sum of several optimized classification algorithms. A class label is obtained by majority voting (hard voting) of individual classifiers [25]. The enumerated class label is then stored in a new class $Cl_{j}$. Here five optimized classifiers such as LGB, CatBoost, XGB, SVC and RF are ensembled to predict COVID-19 accurately, and the block diagram of this ensemble method has been displayed in Figure 2. By way of illustration, XGB, LGB and Catboost provide the “COVID-yes” class (1,1,1), and RF provides the “COVID-no” class. Insight comes from the majority voting system, since most classifiers predict the “COVID-yes” class; therefore, this instant is treated as the “COVID-yes” class. The concept of the ensemble technique is illustrated in the following equation mathematically.
(1)$$\hat{y}=mode\left\{cl_{1},cl_{2},cl_{3},cl_{4},cl_{5}\right\}$$

For instance, if the class labels for LGB, CatBoost, XGB, RF and SVC are calculated as 0, 1, 1, 1, 0, then $\hat{y}=mode\left\{0,1,1,1,0\right\}=1$ because 1 occurs in most of the classes.

### 2.4. Hyperparameter Optimization Using HHO

Hyperparameter optimization plays a crucial role in numerous real-world implementations of ML. The performance of these implementations can be improved by optimizing the hyperparameters of ML algorithms. There are many optimization techniques, such as exhaustive search, gradient descent, genetic algorithms [26], whale optimizer [27], bayesian optimization [28], ant colony optimization [29], etc. Different hyperparameters used by different classifiers are listed as follows: XGB: (*learning_rate*, *colsample_tree*, *gamma*, *max_depth*, *subsample*, *min_child_weight*, *alpha*); LightGBM: (*learning_rate*, *num_leaves*, *feature_fraction*, *bagging_fraction*, *max_depth*, *max_bin*, *subsample*, *colsample_tree*, *min_child_samples*, *min_data_inleaf*); Catboost: (*depth*, *colsample_bylevel*, *subsample*, *n_estimator*, *learning_rate*, *12_leaf_reg*); Random Forest: *min_sample_leaf*, (*max_depth*, *min_sample_split*, *n_estimator*, *criterion*). The appropriate selection (proper tuning) of the values of the hyperparameters can magnify the classification performance. The tuning process can be accomplished in the presence of an optimization algorithm, and this entire procedure is defined as an optimization problem. In our proposed approach, we have formulated a general framework to optimize the hyperparameters of the ML classifiers in Equation (Equation 2).
(2)$$\underset{h\in H}{arg min}f_{obj}\left(Clf\left(H\right);H\right)$$
where h∈H represents the hyperparameters of the classifiers, such as $h_{1},h_{2},h_{3},\cdots ,$$h_{n}\in H$, and Clf is the ML classification algorithms. For the purpose of the optimization task, we need to determine a proper objective function. In this paper, we have proposed an improved objective function, which is defined as an average loss of the K=10-fold cross-validation using F1-score ($f_{obj}$) applied to the training dataset. Because F1-score can be useful even in the case of an imbalanced dataset as it takes both precision and recall into account, it is better than error and accuracy. The mathematical generalization of the proposed objective function ($f_{obj}$) has been illustrated in Equation (Equation 3).
(3)$$f_{obj}=\frac{1}{K}\sum_{k=1}^{K}\left(1-F1Score_{k}\left(Clf\left(H\right)\right)\right)$$
where the $F1Score_{k}$ of a particular classifier, Clf in $k^{th}$-fold can be calculated as,
(4)$$F1Score_{k}=2\frac{\left(Precision\;\times \;Recall\right)}{\left(Precision+Recall\right)}$$

The optimization of hyperparameters is needed to achieve maximum performance on the data in a reasonable amount of time. For our COVID-19 dataset, a metaheuristic algorithm called HHO technique [30] is applied to tune the hyperparameter for precisely detecting COVID-19. The algorithm is designed based on the Harris hawk’s team’s cooperative behaviors of hunting and chasing patterns for the capture of prey in nature called surprise pounce. The prey get confused by the hawks from diversified directions. Harris hawks can efficiently choose chase type in accordance with the distinct patterns of prey [3].

In the proposed HHO-based ML framework for COVID-19 prediction, there are mainly three phases, which have been delineated sequentially as follows:

#### 2.4.1. Exploration Phase

In the exploration phase, according to the concept of the HHO algorithm [31], hyperparameters of the ML classifier are explored for the global optimal position $H_{glo\_opt}$. The search strategies for optimal hyperparameters of ML classifiers can be expressed as:(5)$$H_{i}\left(iter+1\right)=\left\{\begin{array}{cc} H_{rand}\left(iter\right)-r_{1}\left|H_{rand}\left(iter\right)-2r_{2}H_{i}\left(iter\right)\right|, & \mathrm{if}\;Th\geq 0.5\\ {}[H_{glo\_opt}\left(iter\right)-H_{avg}\left(iter\right)]-r_{3}\left[LB+r_{4}\left(UB-LB\right)\right], & \mathrm{if}\;Th<0.5 \end{array}\right.$$
where $H_{i}\left(iter\right)$ and $H_{i}\left(iter+1\right)$ represent the locations of hyperparameters of the classifiers at the present and next iterations (i=1,2,3,⋯,N); *N* denotes the size of population; $H_{glo\_opt}$ symbolizes the position of the global optimal hyperparameter; $H_{rand}$ denotes the randomly selected position of the hyperparameter; iter signifies total iterations executed; LB and UB typify the least and greatest value of each hyperparameter; $Th,r_{1},r_{2},r_{3}$ and $r_{4}$ are randomly selected numeric values between 0 and 1; $H_{avg}$ indicates the mean location of the hyperparameter, that can be calculated as:(6)$$H_{avg}\left(iter\right)=\frac{1}{N}\sum_{iter=1}^{N}H_{i}\left(iter\right)$$

#### 2.4.2. Shifting between Exploration and Exploitation

In the shifting phase, the hyperparameters of the ML classifiers are explored and exploited as flows:(7)$$E=2\times E_{0}\left(1-\frac{iter}{Max\_iter}\right)$$
where $E_{0}$ and *E* represent the initial and escaping energy, respectively. The HHO will be in the exploration phase if |E|≥1 and in the exploitation phase, if |E|<1.

#### 2.4.3. Exploitation Phase

The exploitation phase is stated by defining a random number r∈[0,1]. When 0.5≤|E|<1 and r≥0.5, a soft beiase strategy is used to update the position of the hyperparameter.
(8)$$H_{i}\left(iter+1\right)=\Delta H_{i}\left(iter\right)-E\left|K.H_{glo\_opt}\left(iter\right)-H_{i}\left(iter\right)\right|$$
where *K* is the random number between [0,2]; $\Delta H_{i}\left(iter\right)$ stands for the deviation between the current global optimal position of the hyperparameter and the individual current position, which can be mathematically defined as:(9)$$\Delta H_{i}\left(iter\right)=H_{glo\_opt}\left(iter\right)-H_{i}\left(iter\right)$$

When |E|<0.5 and r≥0.5, a hard beiase strategy [30] is applied to update the position of the hyperparameter.
(10)$$H_{i}\left(iter+1\right)=H_{glo\_opt}\left(iter\right)-E.\Delta H_{i}\left(iter\right)$$

When 0.5≤|E|<1 and r<0.5, a soft beiase strategy [30] with a progressive rapid dive is used to update the position of the hyperparameter.
(11)$$H_{i}\left(iter+1\right)=\left\{\begin{array}{cc} Y, & \mathrm{if}\;f_{obj}\left(Y\right)<f_{obj}\left(H_{i}\left(iter\right)\right)\\ Z, & \mathrm{if}\;f_{obj}\left(Z\right)<f_{obj}\left(H_{i}\left(iter\right)\right) \end{array}\right.$$
where
(12)$$Y=H_{glo\_opt}\left(iter\right)-E\left|K.H_{glo\_opt}\left(iter\right)-H_{i}\left(iter\right)\right|$$
(13)$$Z=Y+S_{rv}\times Levy\left(D\right)$$
where *D* symbolizes the diminution of problem [31]; $S_{rv}$ describes the random vector with size 1×D, where the elements of $S_{rv}$ exist in [0,1]; Levy is the mathematical levy flight function; $f_{obj}\left(*\right)$ is the objective function.

When |E|<0.5 and r<0.5, a soft beiase strategy with a progressive rapid dive is used to update the position of the hyperparameter.
(14)$$H_{i}\left(iter+1\right)=\left\{\begin{array}{cc} Y^{\prime }, & \mathrm{if}\;f_{obj}\left(Y^{\prime }\right)<f_{obj}\left(H_{i}\left(iter\right)\right)\\ Z^{\prime }, & \mathrm{if}\;f_{obj}\left(Z^{\prime }\right)<f_{obj}\left(H_{i}\left(iter\right)\right) \end{array}\right.$$
where
(15)$$Y^{\prime }=H_{glo\_opt}\left(iter\right)-E\left|K.H_{glo\_opt}\left(iter\right)-H_{avg}\left(iter\right)\right|$$
(16)$$Z^{\prime }=Y^{\prime }+S_{rv}\times Levy\left(D\right)$$

By iterating up to maximum iterations (max_iter), the position of the global hyperparameter $H_{glo\_opt}$ is obtained, and using these global optimal hyperparameters, we will establish our proposed ML model for COVID-19 prediction [32].

Using the phases illustrated above, the algorithm of HHO used for the best hyperparameter optimization of classifiers can be constructed as in [Algorithm 1].

### 2.5. Performance Evaluation Metrics

The performance of any classification/prediction algorithm is enumerated using a variety of evaluation metrics. In our proposed approach, the ACC, error, F1-score, SE, SP, MCC, Kappa index, balanced accuracy score, cross-validation score (CV-score), precision and AUC, have been utilized to compute the performance of the proposed research from the confusion matrix. In our HHO-based ML approach, the 2×2 confusion matrix has been used to evaluate the model through the metrics mentioned above. The entries along the principal diagonal represent the correctly classified outcomes. (The higher values of the metrics except the error mentioned above specify the better version of the model.) In addition, the precision-recall curve (PRC) and recall vs. precision boundary are presented to evaluate the performance. Furthermore, a receiver operating characteristic (ROC) has also been used to measure the classification outcomes obtained from the classifiers. The decision boundary threshold value of 0.5 has been considered to provide equal importance to “COVID-yes” and “COVID-no” classes.   
**Algorithm 1:** Proposed HHO-based COVID-19 prediction algorithm
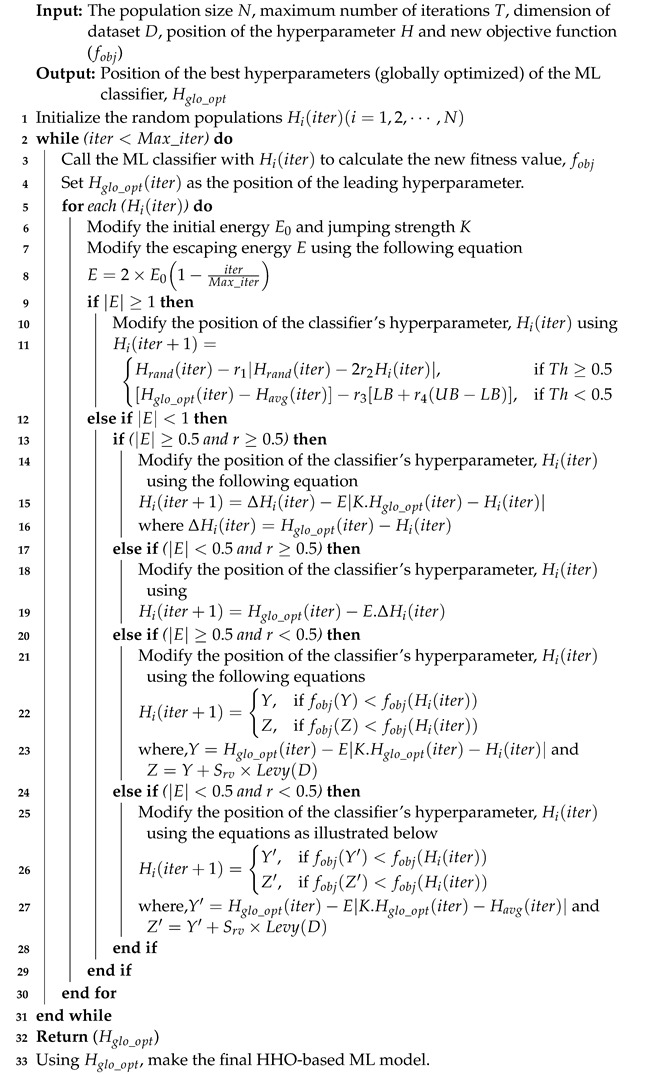


### 2.6. Feature Importance Using SHAP Analysis

SHAP [33], a game theory-based feature analysis technique, briefly delineates how an ML model prognosticates the target variable by computing the contribution of each feature. It uses an additive attribute method where an output model is constructed by linear addition of input variables. The linear function f(z) is defined by the additive attribute method.
(17)$$f\left(z\right)=\psi_{0}+\sum_{j=1}^{N}\psi_{j}z_{j}^{{}^{\prime }}$$
where *M* and *N* denote the number of input features and set of input features, respectively; *z* typifies the training data; $z^{\prime }\in {\left\{0,1\right\}}^{M}$ represents the coalition vector (simplified features), where “1” and “0” denote the presence and absence of the corresponding features; $\psi_{j}\in R$ is the feature attribution for a feature *j*. The contribution of each feature can be allocated by obtaining $\psi_{j}$ from the following equation.
(18)$$\psi_{j}=\sum_{S_{ftr}\in N\backslash j}\frac{|S_{ftr}\left|!(n-\right|S_{ftr}\left|-1\right)!}{n!}\left[f_{x}\left(S_{ftr}\cup \left\{j\right\}\right)-f_{x}\left(S_{ftr}\right)\right],$$
where $S_{ftr}$ is the subset of features, *n* is the total input features.

## 3. Experimental Result

This paper uses HHO to optimize the hyperparameter of five classification algorithms. The optimized values of the hyperparameters using HHO and their domain range are shown in Table 2. For a fair justification, this study used the same dataset for the same time frame with the same set of training and testing samples in all classifiers. Different statistical measures were calculated to estimate how our proposed ML model performs. Table 3 shows different performance measures of the COVID-19 dataset. This section depicts the precision versus recall curve and the recall rate versus decision boundary. Furthermore, the ROC curve has been presented to show the performance. Moreover, the feature analysis using SHAP has been presented in Section 3.5. In Section 4.1, a comparison with the previously accomplished related works has been provided [as shown in Table 4]. Finally, a potential application of our HHO-based framework has been outlined through DSS in Section 5.

### 3.1. HHO Result

Table 3 shows that the ensemble method picks the peak point in the case of all classification metrics such as ACC (92.29%), CV-score (92.68%), balanced ACC (92.68%) and AUC (97.80%) with a very negligible error (7.32%), whereas the HHOXGB occupies the second-highest position. On the other hand, the average performance of the BernoulliNB, LDA and QLDA was very low for the COVID-19 dataset in terms of all classification measures. The (69.63%) accuracy of QLDA was the lowest compared to other classifiers considered in this proposed research. To show the effectiveness of HHO, an experiment has also been performed without HHO using SVC, Catboost, RF, XGB and LGB. It has been evidenced that all the classifiers except Catboost provided a lower performance in all classification metrics when HHO was not been utilized. For instance, ACC (83.16%), Cohen Kappa (66.31%) and Precision (83.16%) values of XGB without HHO remain at a lower value than that of the HHO-optimized XGB. The code without HHO is also provided in the Github repository; https://github.com/MIrazul29/HHO-ML (accessed on 18 March 2022).

### 3.2. Precision vs. Recall Curve

The PRC is generally utilized to get a clear classification overview that helps evaluate and compare the test result. The more the curve adjacent to the right corner of the graph is, the more the test result is perfect. Figure 3 shows that the PRC value is very close to the right corner of the curve, implying that the ensemble method provides superior performance, while NB stays at the lowermost point.

### 3.3. Recall vs. Decision Boundary

The recall rate has been enumerated in our proposed classification task depending on a fixed threshold (*T*). To exemplify, in Figure 4, 0.5 threshold value for “COVID-yes” class has been considered to demystify the relationship between the recall rate (displayed along the *y*-axis) and the decision boundary (visualized along the *x*-axis), where the best recall rate has been obtained from the KNN provided, 0.9268, implying that approximately 92.68% of the time, KNN accurately classifies the “COVID-yes” class, whereas HHOXGB correctly classifies the recall rate for “COVID-no”.

### 3.4. ROC Curve

The probability/performance of a binary outcome while classifying or predicting is defined as the ROC curve. The graphical representation of the ROC curve in Figure 5 shows that the ACC of the ensemble method grabs the apex in terms of ACC, whereas the LDA remains at the lowest point.

### 3.5. Feature Importance (SHAP Value Analysis)

Feature importance represents how the input features are convenient in predicting a target variable. To accomplish this task, the technique of feature importance assigns a score to the input features and provides insights into the predictive model (classification model). Most importantly, the enumeration of feature importance upgrades the effectiveness and efficiency of the predictive modeling project. We have used the SHAP summary plot in this study. The use of the SHAP summary plot (Figure 6) has two-fold advantages as we can see: feature ranking and the effect of each feature. The position in the *y*-axis determines the feature ranking in descending order (higher importance to lower importance). The effect of each feature is determined by *x*-axis SHAP values; positive SAHP values indicate a positive correlation with the target and vice versa. In addition, the feature value indicated by the red color represents a higher feature value, and blue indicates a lower feature value. The jagged overlapping points make sense of distribution.

The significance of the features for any classification or prediction can be effortlessly analyzed by sorting the features in plunging mode, where the most influential feature occupies the peak point. For instance, as manifested in Figure 6, the bar plot visualizes the top 20 features, where “*COVID19_contact*” is in the topmost position. The next dominating features are “*COVID19_symptoms*”, “*cannabis*”, “*bmi*”, “*age*”, “*contact_counts*”, etc. In contrast, “*hiv_positive*” remains undersurface compared to the other features exhibited in the given diagram. Moreover, from Figure 6, it can be seen that higher “*COVID-19_contact*”, “*COVID19_symptom*”and “*health workers*” show a negative SHAP value, i.e., negative correlation. It intuitively reveals that the higher values of this feature indicate less chance to survive, i.e., COVID-no in this case and vice versa. Note that the summary plot is a bird’s eye view of the data. The SHAP dependence plot and SHAP feature interaction plot can be used to discuss a particular feature and particular instance. This may determine the effect of one specific feature in model performance improvement, which is beyond the scope of this paper, and we will keep it in our near future works.

## 4. Discussion

In this research, we have used the HHO technique, a metaheuristic optimization algorithm, to tune the hyperparameters of some state-of-the-art classifiers, including HHORF, HHOXGB, HHOCAT, HHOLGB and ensemble methods, and thus amplify the classification ACC. To enumerate the performance of the COVID-19 prediction task, we have considered several classification measures, such as ACC (92.68%), AUC (97.80%), error (7.32%), etc. In addition, some pictorial delineations, such as the PRC, recall vs. decision boundary curve, and the ROC curve, have been generated to visualize and prove the superiority of the performance of the proposed method. It is elucidated that the ensemble method occupied the peak point in terms of all classification measures. In contrast, HHOXGB, another state-of-the-art classification algorithm, captured the second-highest position. In the SHAP analysis (Figure 6), the mean SHAP values, representing the average impact on model output magnitude, have been enumerated to select the most dominating features, where “*COVID19_contact*” was the most influential over the rest of the features existing in the dataset used.

It can be further added that our proposed approach is likely to be implemented not only for predicting COVID-19 patients but also for detecting numerous diseases, such as asthma, vaccination [34], hypertension [35], diabetes mellitus [36], etc. The integration of the moderately large-scale dataset and utilization of an independent dataset to validate the proposed framework before further clinical trials will tremendously intensify the feasibility of this research.

### 4.1. Comparative Study

The performance of our proposed study is superior to the other existing methodologies, which has been clarified by a comparative study illustrated in Table 4. It is evidenced by Jim et al. [37] who considered the concept of CNN and received an ACC of 92.50%. In addition, He et al. [38], Ahamad et al. [39] and Li et al. [40] implemented XGB on clinical data and obtained average ACC of 87%, which outperforms the ACC obtained by Brinati et al. [41], who simultaneously used DT and RF. Moreover, Chimmula and Zhang [42] employed an LSTM network and attained an ACC of 92.67%. From the detailed delineation mentioned above, it is clear that our proposed framework provides the best result in terms of all classification measures. Note that in Table 4, we have relied on the results published as the datasets used in those studies are not publicly available. This is a common as well as a complex problem. Even the results will be different on the same publicly available dataset due to different training and testing samples. Therefore, we have used big data and provided processed training and testing datasets as well as code on GitHub for reproducibility and future contributions and collaboration. Moreover, comparative studies presented in Table 4 used the same state-of-the-art classifiers we used. Therefore, from the classifier’s point of view, the HHO-based machine learning model ensures the optimal use of state-of-the-art machine learning algorithms and improves performance.

**Table 4 diagnostics-12-01023-t004:** Comparison with related works.

References	Classifiers	Dataset Used	ACC	SE	SP	AUC
Jim et al. [37]	Deep Convolutional Neural Network	Clinical Image Data	92.5%	94.2%	95.6%	
He et al. [38]	XGB	Clinical, Blood samples of 75 Features	90%			
Ahamad et al. [39]	XGB, RF, DT, SVM	Demographic and Symptom	85%	90%		
Li et al. [40]	XGB	Clinical Data		92.5%	97.9%	>90%
Brinati et al. [41]	DT, RF	Hematochemical Values from Blood Exams	86%	95%		
Chimmula and Zhang [42]	Deep learning using LSTM	Demographic	92.67%			
**Proposed**	Ensemble Method	Clinical Data	92.68%	92.68%	92.68%	97.80%

## 5. Potential Application

The potential application of the proposed model can be effortlessly obtained through designing a decision support system (DSS). A DSS is a pictorial interpretation to display the probable state of the labels of the target variable. In our proposed HHO-based COVID-19 prediction model, a clinically executable DSS has been designed to portray the probable state of “COVID-yes”; in other words, it represents whether a patient is carrying “COVID-yes” or not in a probabilistic sense.

Note that the threshold value of 0.5, i.e., probability = 0.5, has been considered according to the general rule of machine learning. If the threshold value is less than or greater than 0.5, then there may arise Type-1 and Type-2 errors. Moreover, probability = 0.5 is the middle point between 0 and 1, which provides a balanced and unbiased calculation of sensitivity, specificity and other performance metrics. If the probability calculated by the ensemble method is greater than 0.5, it represents that a patient would have a higher chance of “COVID-yes”. This graphical representation will benefit nonspecialist users such as clinical staff and nurses. The previously calculated probability from the ensemble method has been exercised to visualize an expected output of “COVID-yes” as represented in Figure 7. It is worth noting that in the upper figure in our predicted label is highly accurate to the true label except for the specific patient (marked by the red rectangle in Figure 7). Moreover, the corresponding probability of “COVID-yes” has been shown in the lower figure. The test data are sorted in escalating mode to initially visualize the “COVID-no” class and then the “COVID-yes” class.

## 6. Conclusions

This paper designs an ML framework to predict COVID-19 using a publicly available clinical dataset. Five up-to-the-minute ML classification algorithms have been applied, where the hyperparameters of each classifier have been optimized using the proposed HHO algorithm with an improved objective function. The proposed approach aims to work in a real-time inpatient facility dataset so that the method can be user-friendly and cost-effective. The outcome of the proposed system has been evaluated by some metrics such as ACC, F1-score, MCC, Kappa index, etc. A DSS has been created to display the probability of a COVID-19 attack as a potential application of our proposed system. This can be very useful from the patient end as well as from a clinical point of view. The clinical staff and caregivers can easily screen COVID-19 patients. Furthermore, the proposed optimization technique, applied to optimize the hyperparameters of the ML classifiers, can easily be adapted for other disease predictions such as diabetes, asthma and hypertension. The feasibility of the clinical trials of the proposed research can be ensured by implementing the proposed framework on a larger dataset. Last but not least, our research would likely be integrated into mobile devices for the usefulness of end users.

## Figures and Tables

**Figure 1 diagnostics-12-01023-f001:**
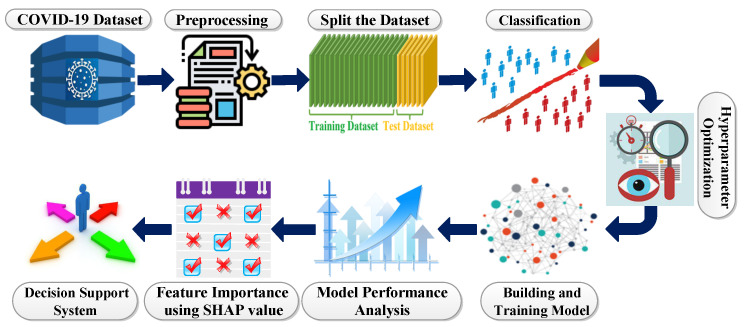
Proposed system architecture.

**Figure 2 diagnostics-12-01023-f002:**
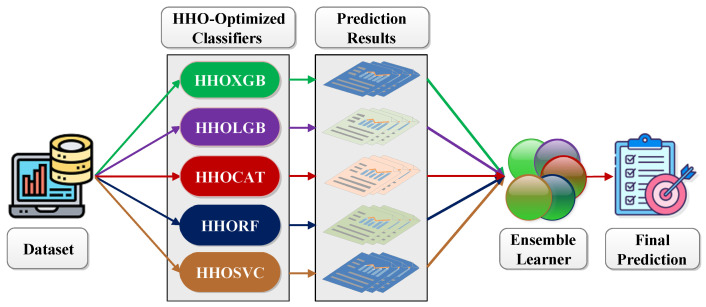
Block diagram of ensemble method.

**Figure 3 diagnostics-12-01023-f003:**
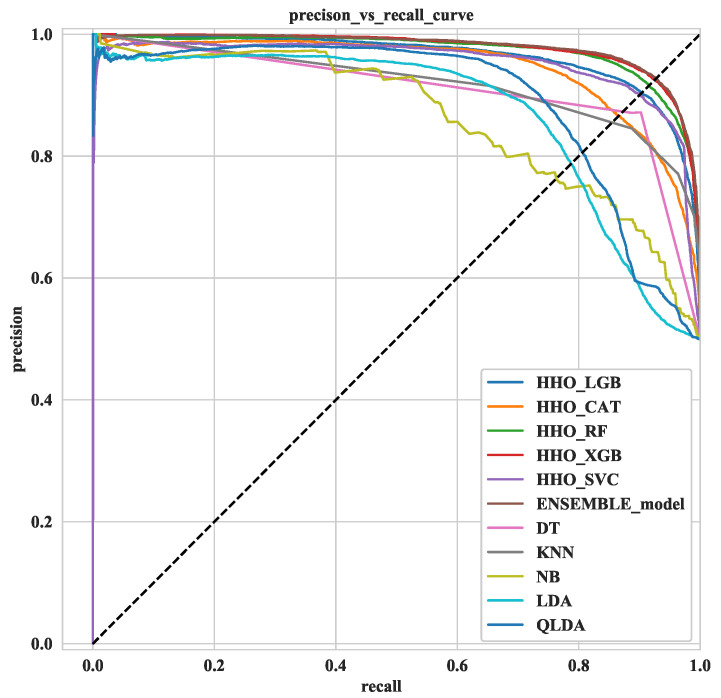
Precision vs. recall curve.

**Figure 4 diagnostics-12-01023-f004:**
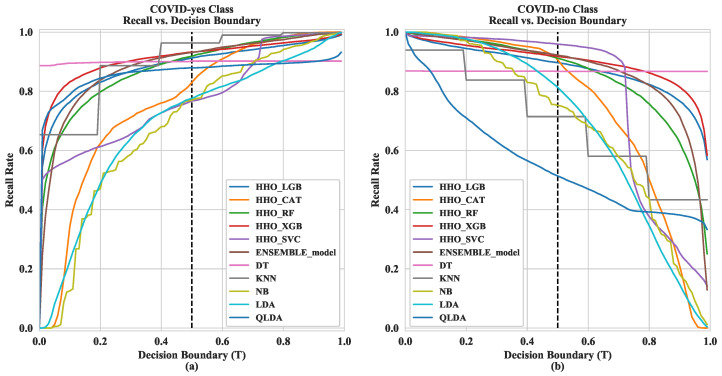
Visualization of recall vs. decision boundary for (**a**) “COVID-yes” class and (**b**) “COVID-no” class.

**Figure 5 diagnostics-12-01023-f005:**
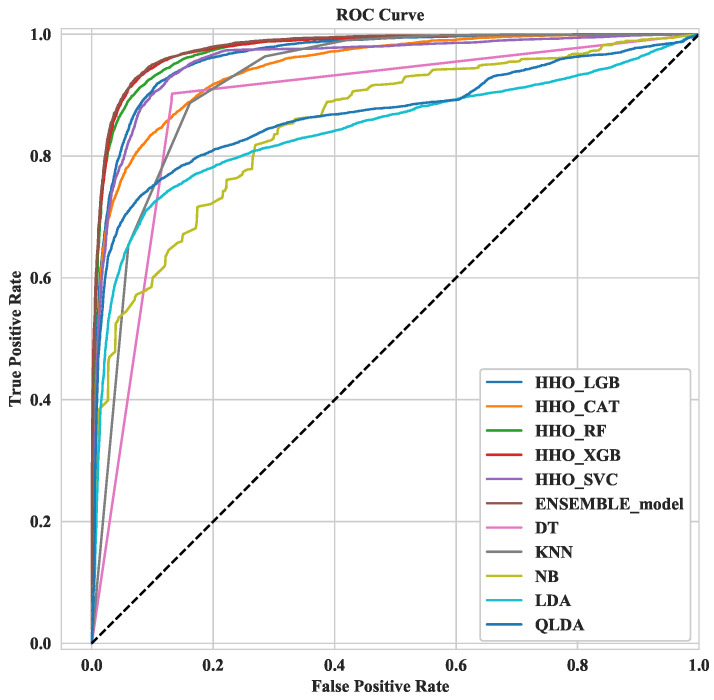
ROC Curve.

**Figure 6 diagnostics-12-01023-f006:**
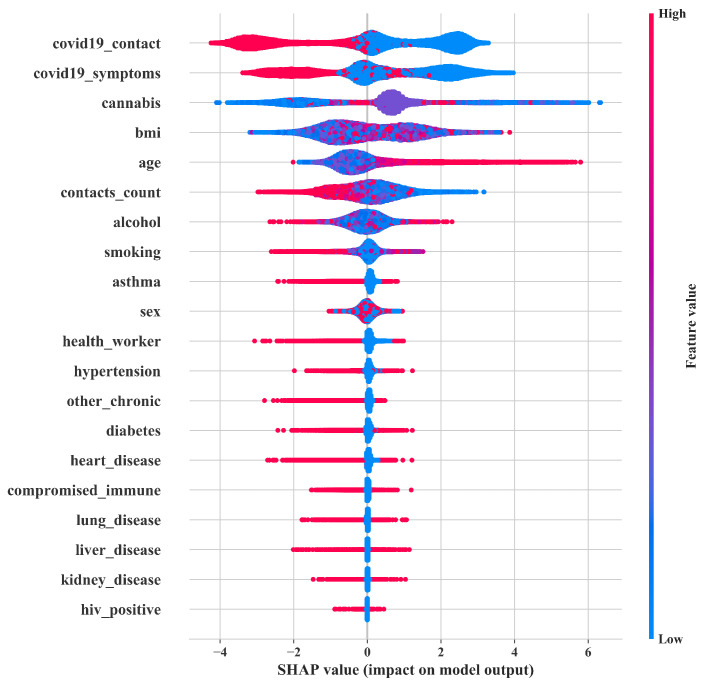
SHAP analysis.

**Figure 7 diagnostics-12-01023-f007:**
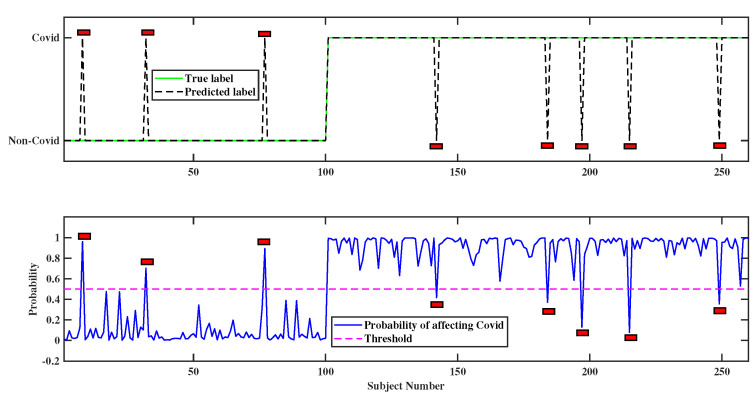
Development of a DSS using a HHO–based technique.

**Table 1 diagnostics-12-01023-t001:** Description of the dataset used for COVID-19 prediction.

Variable Name	Description	Variable Type	Ratio of Boolean, 0/1	Missing Value (Percentage)
BMI	Measures a healthy connection between the weight and height of a person	numerical		
Alcohol	An ethanol-based organic compound used in manufacturing different drugs	numerical		0.2%
Cannabis	A psychoactive drug made from the cannabis plant which is also called marijuana	numerical		28%
Contacts-count	Defines number of individuals contact with the patients	numerical		0.4%
Age	Defines the number of years of a person’s existence	categorical		
COVID-19 Symptoms	Symptoms that can be noticed if a person is COVID-19 positive	categorical	999899/23527	
COVID-19 Contact	Defines number of contact with COVID-19 positive individuals	categorical	973763/49663	
Asthma	A persistent lung disease which causes breathing difficulties	categorical	867913/1555	
Kidney-disease	Kidney conditions that create inconvenience while filtering blood in the body	categorical	1019551/3875	
Liver-disease	Diseases that hamper the regular functions of a liver and cause significant damages	categorical	1020832/2594	
Compromised-immune	Having a weak immune system with a chance of getting infected by various diseases easily	categorical	965426/58000	
Heart-disease	Unfavorable heart conditions or diseases of an unhealthy heart	categorical	1003713/19713	
Lung-disease	Some breathing disorders and diseases that affect one’s lung	categorical	1008154/15272	
Diabetes	A disease that occurs when one’s body becomes unable to maintain insulin and the blood sugar becomes high	categorical	959537/63889	
HIV-positive	A physical state when a person’s body contains fully functional HIV virus and most likely have AIDS	categorical	949839/73587	
Hypertension	High blood pressure due to several health issues and other circumstances	categorical	879542/1438	
Other-chronic	Persistent and long-lasting health conditions	categorical	949839/73587	
Health-worker	A person who works for health-related issues and provide basic healthcare	categorical	1002710/20716	
Sex	Expresses gender differences among humans	categorical	63389/382573	
Smoking	Inhaling smoke from burnt plant material	categorical		0.02%
COVID-19_positive	A physical state when one’s body contains fully functional COVID-19 virus	categorical	1011257/12169	

**Table 2 diagnostics-12-01023-t002:** Tune Hyperparameters and their domain range.

Classifiers	No. of Hyperparameter	Hyperparameters	Domain Range
Extreme Gradient Boosting	7	*Learning rate*	($1.5\times 10^{-15}$ – 0.9)
		*Colsample_tree*	(0.001–1.00)
		*Min_child_weight*	(1–200)
		*Gamma*	($1\times 10^{-9}$ – 1.0)
		*Subsample*	(0.001–1.0)
		*Max_depth*	(1–200)
		*Alpha*	($1\times 10^{-6}$ – 1.0)
Light Gradient Boosting	10	*Learning_rate*	(0.01–1)
		*Max_bin*	(15–100)
		*Num_leaves*	(20–100)
		*Bagging_fraction*	(0.6–1.0)
		*Feature_fraction*	(0.1–0.9)
		*Max_depth*	(5–50)
		*Subsample*	(0.1–1.0)
		*Colsample_tree*	(0.01–1.0)
		*Min_child_samples*	(3–100)
		*Min_data_inleaf*	(90–120)
CatBoost	6	*Depth*	(1–12)
		*Colsample_bylevel*	(0.01–0.1)
		*Subsample*	(0.01–0.1)
		*n_estimator*	(100–400)
		*Learning_rate*	(0.001–0.01)
		*12_leaf_reg*	(1–9)
Random Forrest	5	*Min_sample_split*	(1,20)
		*Min_sample_leaf*	(1,20)
		*n_estimator*	(10,1000)
		*Criterion*	(“Gini,” “entropy”)
Support Vector Classifier	2	*Cost*	(0.001,20)
		*Gamma*	(−6,6)

**Table 3 diagnostics-12-01023-t003:** Classification performance on COVID-19 dataset.

Classifiers	Performance Indexes												
	CV-Score	AC	Err	F1-Score	FPR	Kappa	MCC	PPV	SEN	SPE	Threat-Score	BAC	AUC
**HHOLGB**	90.13%	90.47%	9.53%	90.47%	9.53%	80.94%	80.95%	90.48%	90.47%	90.47%	82.60%	90.47%	96.40%
**HHOCAT**	86.87%	86.81%	13.19%	86.79%	13.20%	73.62%	73.83%	87.02%	86.81%	86.80%	76.67%	86.81%	94.50%
**HHORF**	91.68%	91.53%	8.47%	91.53%	8.47%	83.06%	83.06%	91.53%	91.53%	91.53%	84.38%	91.53%	97.40%
**HHOXGB**	92.23%	92.54%	7.46%	92.54%	7.46%	85.09%	85.10%	92.55%	92.54%	92.54%	86.12%	92.54%	97.70%
**HHOSVC**	83.50%	84.54%	15.46%	84.29%	15.48%	69.07%	71.33%	86.83%	84.54%	84.52%	72.89%	84.53%	95.60%
**ENSEMBLE_MODEL**	**92.38%**	**92.67%**	**7.33%**	**92.67%**	**7.33%**	**85.34%**	**85.35%**	**92.67%**	**92.67%**	**92.67%**	**86.34%**	**92.67%**	**97.80%**
**DT**	87.71%	88.43%	11.57%	88.42%	11.57%	76.85%	76.89%	88.47%	88.43%	88.43%	79.25%	88.43%	88.50%
**KNN**	83.93%	83.89%	16.11%	83.64%	16.09%	67.80%	70.01%	86.15%	83.89%	83.91%	71.94%	83.90%	92.20%
**BernoulliNB**	76.61%	76.11%	23.89%	76.11%	23.89%	52.22%	52.22%	76.12%	76.11%	76.11%	61.43%	76.11%	85.20%
**LDA**	79.39%	79.50%	20.50%	79.49%	20.50%	59.00%	59.04%	79.54%	79.50%	79.50%	65.97%	79.50%	84.30%
**QLDA**	70.01%	69.63%	30.37%	68.59%	30.34%	39.29%	42.22%	72.69%	69.63%	69.66%	52.49%	69.64%	86.90%
**SVC (without HHO)**	83.87%	83.86%	16.14%	83.80%	16.15%	67.71%	68.21%	84.36%	83.86%	83.85%	72.13%	83.85%	87.30%
**RF (without HHO)**	86.22%	86.63%	13.37%	86.63%	13.36%	73.27%	73.32%	86.69%	86.63%	86.64%	76.41%	86.63%	86.70%
**LGB (without HHO)**	86.56%	86.77%	13.23%	86.77%	13.23%	73.54%	73.55%	86.78%	86.77%	86.77%	76.63%	86.77%	93.90%
**XGB (without HHO)**	83.38%	83.16%	16.84%	83.15%	16.84%	66.31%	66.32%	83.16%	83.16%	83.16%	71.17%	83.16%	90.10%
**CAT (without HHO)**	86.90%	86.94%	13.06%	86.92%	13.07%	73.87%	74.10%	87.16%	86.94%	86.93%	76.86%	86.93%	94.60%

## Data Availability

The processed data, trained model, and codes related to this study are available at: https://github.com/MIrazul29/HHO-ML (accessed on 18 March 2022).

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
