# Peer review of "An Improved Machine-Learning Approach for COVID-19 Prediction Using Harris Hawks Optimization and Feature Analysis Using SHAP"

_diagnostics, 2022, doi:10.3390/diagnostics12051023_

Round 1

Reviewer 1 Report

This paper proposes an ML framework leveraging Harris Hawks Optimization algorithm with an improved objective function applied to optimize the hyper-parameters of multiple ML algorithms including XGB, LGB, CAT, RF, and SVC. An ensemble model was then produced by applying these optimized ML algorithms. The proposed method has been applied to a publicly available COVID-19 dataset and obtained 92.38% accuracy. The authors have also described the existing works in detail in the Related Works section. 

Though the problem space this paper is trying to address is very interesting, the paper lacks some crucial elements, highlighted in the comments section.

Comments:

  1. The authors mention that the proposed model uses inpatient facility data instead of X-ray or CT scan images and that people do not require to go to the hospital to test COVID-19. This sounds contradictory, and a proper description of this would help.
  2. Typo: COVID-19 (be consistent with upper/lower case)
  3. The dataset contains 1023426 records. Does each of these belong to different individuals?
  4. The dataset is highly skewed. How the authors dealt with class-imbalance is not explained clearly.
  5. Line 132: Figure 1 does "not" show the columns with a higher rate of missing value than others. Is this a typo?
  6. The reasoning behind how the authors determined risk_infection/risk_mortality are not important to the analysis is not mentioned.
  7. Line 145-146: Rephrase the sentence. 
  8. Line 130: What iterative imputation techniques were used? What was the estimator?
  9. A lot of content, for example, sections 2.3 and 2.6 - does not provide new information. These concepts already exist, and need not be explained much in detail.
  10. When citing a concept/algorithm, please make sure to cite legit publications (paper/book/..) rather than random websites.
  11. Line 271: 'cashing' - typo
  12. Was the work compared with other objective functions?
  13. Figure 2: Are 'Optimized Classifiers' an input to the already optimized ML algorithms? Or is it the inconsistent use of input/output arrows in the figure?
  14. Sections 2.4.1-2.4.3: Please add citations as you see fit.
  15. Algorithm 1: Use the equations in-place. Referring to the text inside the algorithm is not the correct way to write an algorithm.
  16. Table 3: A proper comparison would be to include with HHO vs. without HHO for LGB, XGB, CAT, RF, and SVC. 
  17. Section 3.5: There's no detailed discussion (neither in section 4) on the conclusion from SHAP values. For example, whether the feature 'health worker' is an important feature, and what is the conclusion from SHAP values. Also, were there any improvement in terms of feature usage in the model after looking at the SHAP values?
  18. Section 4.1: Did the study use the same dataset for the same timeframe, with the same set of samples? If not, the comparison of accuracy is invalid. Same with Table 4.
  19. Figure 7: How did you arrive at the threshold?
  20. Citation 4: Typo: Who --> WHO

Author Response

 We want to thank Reviewer 1 for your constructive comments, which have
significantly improved technical content and presentation quality.
 Please find the attached reviewer response.

Reviewer 2 Report

Does the introduction provide sufficient background and include all relevant references?
The Introduction section of the manuscript provides the theoretical background of the research. The manuscript provides a good summary of relevant and recent literature related to the topic.

Is the research design appropriate?
In this manuscript, design research aims to combine relevance to the purpose of research with innovation in the procedure. For these reasons, it appears to be appropriate, clear and engaging from the first reading of the manuscript.

Are the methods adequately described?
The Materials and Methods section is described comprehensively and straightforwardly. 

Are the results clearly presented?
In the Results section, the results of the analysis are well explained both the tables and the figures simply represent the results.

Are the conclusions supported by the results?
The manuscript provides essential conclusions and implications.
The conclusions are supported by the results obtained, which are widely discussed both quantitatively and qualitatively.  

Author Response

 We want to thank Reviewer 2 for your constructive comments, which have
significantly improved technical content and presentation quality.
 We also appreciate your positive feedback. Please find the attached reviewer response.

Round 2

Reviewer 1 Report

The authors have addressed several of the comments mentioned in the previous review.

Comments/Suggestions:

  1. Lines 107-109: Typo: Still lacks clarity.
  2. Line 171: rephrase correctly
  3. Figure 2: The blue arrows are still confusing, and makes it look like they are inputs.
  4. Line 276: mean of the K-fold cross validation...
  5. Algorithm 1: An algorithm should be self-contained, and not draw references to the text.
  6. Table 3 and section 3.1: Still lacks the results of all classifiers without HHO (eg., RF, LGB, etc). It is also important to capture all of the metrics for classifiers without HHO for comparison of classifiers with HHO.
  7. Are sections 2.4.1 and 2.4.2 the authors' own contributions? If not this section should be made as concise as possible. 

Author Response

Dear Reviewer

We want to thank you for your constructive comments, which have significantly improved technical content and presentation quality. Furthermore, we have taken into consideration all comments and suggestions in the revised manuscript. In the response letter, we have also described how we have addressed all comments and the changes that we have made. Please see the attached file and revised manuscript.

Round 3

Reviewer 1 Report

The authors have addressed the crucial suggestions/comments in this version.